# Intimate Partner Violence and Resilience: The Experience of Women in Mother-Child Assisted Living Centers

**DOI:** 10.3390/ijerph17228318

**Published:** 2020-11-10

**Authors:** Chiara Rollero, Federica Speranza

**Affiliations:** Department of Psychology, University of Turin, 10124 Torino, Italy; fedespera@libero.it

**Keywords:** intimate partner violence, resilience, Mother–Child Assisted Living Center

## Abstract

Research has largely documented the damaging consequences of intimate partner violence. However, the literature presents an important gap in the identification of factors that may strengthen resilience in the victims, especially in the case of mothers and pregnant women. The present study aimed at investigating the experience of abused mothers engaged in an educative path in a Mother–Child Assisted Living Center. A qualitative descriptive methodology was used. Face-to-face in-depth interviews were conducted with a purposive sample of eight women. Four main themes emerged from the interviews: (1) improvement in the mother–child relationship; (2) a process of personal change during the educative path; (3) the rebuilding of trust relationships; and (4) attitudes and hopes toward the future. Taken together, these findings highlight the process of resilience, conceived from a socioecological perspective as the ability to use resources rooted in interconnected systems. The implications of these findings are discussed.

## 1. Introduction

Intimate partner violence (IPV) is an abuse of power committed by a romantic partner in a relationship or after separation. It takes several forms, such as emotional abuse, physical and/or sexual violence, intimidation and threats, and social or economic deprivation [1].

In the United States, it is estimated that approximately one in four women have been victims of sexual violence, physical violence, and/or stalking from their partner in their lifetime [2]. Similarly, 1 in 10 women has experienced some forms of IPV across the 28 States of the European Union [3]. For its prevalence and its physical, psychological, social, and economic consequences, intimate partner violence is considered a salient public health issue [4].

Literature has clearly shown that IPV has a large amount of negative physical and psychological consequences for the victims. Evidently, IPV puts women at risk of experiencing body injuries, such as contusions, broken bones, and life-threatening wounds [2,5]. Physical health consequences also comprise neurologic dysfunction, cardiovascular problems, somatic symptoms, and compromised sexual and reproductive health [2,5,6,7]. Concerning psychological functioning, victims of IPV are at increased risk of post-traumatic stress symptoms, depression, and anxiety [2,5,6]. While all women may experience such poor health consequences, IPV-exposed pregnant women and mothers are at increased risk of vaginal bleeding, edema, high blood pressure, and more frequent hospital visits during pregnancy [8]. Furthermore, pregnant women exposed to proximal or distal IPV show elevated depressive symptoms, both during pregnancy and postpartum [9,10,11]. Additionally, mothers of young children show higher risk for suicidal ideation [12], self-harming thoughts [13], and higher stress levels [14].

Exposure to IPV may also increase parenting stress and foster risky behaviors, compromising healthy child development. According to the “spillover hypothesis”, conflict or lack of support in partner relationships may extend beyond this dyad and impede good parenting by increasing stress and dysfunctional interactions [15,16]. Consistently, research has found that maternal reports of IPV are significantly linked to parenting stress and unsupportive parenting [17,18], and parenting stress mediates the effect of IPV on children’s behavioral difficulties [18]. Furthermore, exposure to IPV may create painful memories that decrease positive attitudes and emotions about motherhood, alter the mental representation of children, and disturb an appropriate mother–child bonding [19,20,21].

When children development is considered, the literature has shown that maternal experiences of IPV are linked to poorer motor and language abilities in children [22]. In a recent study carried out in eleven countries with 15,000 mothers of children aged 36 to 59 months, Jeong and colleagues [23] found that exposure to IPV was negatively linked to early child development (assessed as cognitive, literacy, numeracy, socioemotional, and physical development). This negative association was partially explained by low levels of stimulation, i.e., activities such as telling stories, singing songs, or drawing together, that a mother engages with her child.

Other studies have found that maternal IPV exposure may have long-term consequences that last throughout the childhood years. IPV-exposed youth are more likely to show adjustment difficulties, substance abuse, low self-esteem, posttraumatic stress disorder, emotional problems, and depressive symptoms [24,25]. They can also experience more frequent academic, cognitive, and social difficulties, as well as behavioral struggles [24,26]. Thus, addressing IPV as early as possible is critical not only to protect victims, but also to potentially preserve children from the damaging outcomes of exposure to violence [5].

### 1.1. IPV and Resilience

If the literature has largely documented the detrimental effects of IPV, few attention has been dedicated to victims able to overcome the adversity and emerge with new perspectives. According to McNaughton and colleagues [27], the literature presents an important gap in reference to the identification of factors that may buffer the impact of IPV and strengthen resilience in victims, especially in the case of mothers and pregnant women.

Resilience is the ability to overcome adversity, resulting from the interplay between risk and protective factors, rooted in interconnected systems, such as the individual, family, community, and culture [28]. From a socioecological perspective, “resilience is the ability of individuals (on their own and collectively) to navigate the culturally relevant resources they need to do well when confronting adversity, as well as their capacity to negotiate for these resources to be provided in ways that are meaningful” ([29], p. 40).

One study carried out with battered women living in shelters found that resilience was associated with determination, self-pride, a lack of dependency on other’s opinions, support systems, and ability to mobilize resources [30]. Another research [31] investigated battered women’s narratives and showed the importance of a new self-attitude, contempt for their relationships, and internal changes. Consistently, in their recent study with victims of IPV, Brosi and colleagues [32] found that women were more likely to show resilience and posttraumatic growth when they changed their life perspective, had access to social support, and were motivated to end the cycle of violence for their children.

Other research has considered the effects of specific programs targeted at victims of IPV. Easterbrooks and colleagues [33] used longitudinal data from a sample of U.S. young mothers and demonstrated that the effects of a home visiting program on maternal well-being (both parenting stress and risky behaviors) two years after program enrollment were mediated by reductions in IPV one year earlier. Another study assessed the outcomes of the Nurse–Family Partnership program for pregnant women [34]. In this program, women received home visits by trained nurses during pregnancy and the first two years of their child’s life. The aims of such visits were to provide housing assistance, to empower women to become financially independent, and to foster supported healthy emotional regulation strategies between partners. At 32 weeks of pregnancy, compared to the control group, women in the intervention program reported fewer incidents of psychological aggression, physical assault, and sexual coercion. After two years, only physical assault remained lower for the intervention group [34]. According to Howell and colleagues [5], the Nurse–Family Partnership represents a notable program for its grounding in human ecology and social perspectives but it may have only limited effectiveness because it does not include a specific protocol of IPV intervention.

### 1.2. The Current Study

In line with literature underlining the necessity to explore resilience among abused women [27], the present study aimed at advancing the existing knowledge about victims of IPV able to overcome the adversity. As seen, the literature presents an important gap on this topic. Thus, the main goal was investigating the experience of abused mothers engaged in an educative path in a Mother–Child Assisted Living Center, with a specific focus on indicators of resilience during such path. Since we were interested in women’s description and the interpretation of their own situation, a qualitative methodology was selected.

## 2. Materials and Methods

### 2.1. Participants and Procedure

The Ethics Committee of the University of Turin, Italy, approved the study protocol (ethical approval code CERP 15731). Participants were recruited through four Mother–Child Assisted Living Centers located in the North of Italy. The Mother–Child Assisted Living Centers are structures aimed at hosting mothers and their children experiencing difficult situations. In most cases, mothers are victims of abuse, IPV, other forms of violence, and/or are facing a trauma and are not able to adequately take care of themselves and their children. The Centers provide a safe environment and should represent the first step in the process of overcoming adversity and gaining independence. They are managed by a multi-professional team, which includes educators, psychologists, childcare assistants, and cultural-linguistic mediators. The team elaborates a specific educative path for each family unit, whose goal is accompanying women toward autonomy and full social reintegration. Usually, the educative path lasts from six months to two years. Each Center can host from 8 to 14 individuals, including children aged 0–16 years.

In each Center, potential participants were presented the study, its aim and duration, and data treatment. Consenting women were assured that they could discontinue the study at any time. They were also informed that all information provided by them would be de-identified to protect confidentiality. Written informed consent was obtained from each participant. Their involvement was completely voluntary: no benefit was offered in exchange for participation.

Eight women participated in the study. All of them had been victims of IPV, and four of them were still living in a Center. Their age range was 19–42 years. Three were Italian, two were from Nigeria, one from Peru, one from Romania, and one from Morocco. Further information concerning their children and the duration of the educative path in the Center is presented in Table 1.

### 2.2. Data Collection

Data were collected by means of face-to-face in-depth interviews between September and December 2019. They were carried out in Italian by one of the female authors who was familiar with the Centers, as she did her internship there. Citations reported here were translated into English. The interview schedule began with brief socio-demographic and introductory questions and then addressed the experience of the women during their treatment at the mother–child assisted living Center. When necessary, the researcher used prompt questions about their relationship with the children, changes in self-awareness, autonomy development, and attitudes and opinions about the educative path in general. Specifically, the most frequent prompt questions were: “Did the relationship with your child change?”, “Do you think you have changed as a woman?”, “How did this educational path affect the way you feel toward yourself?”, “What is your opinion about the educational path in general?”, “Did you learn something important to you?”. The questions were developed by the authors and were pre-tested in two pilot interviews with two women engaged in the educative path in a Center.

The interviews lasted between 30 min and 1 h and 10 min, with an average length of 40 min. They were audiotaped and then transcribed verbatim for analysis. Verbatim transcription implies the word-for-word reproduction of verbal data, so that the written words are an exact replication of the audiotaped words [35].

### 2.3. Data Analysis

A thematic analysis was carried out without a predetermined coding scheme, following Corbin and Strauss’ guidelines [36]. The flexibility of this approach, along with the well-established guidelines for carrying out the analysis, informed the choice of the method. Such analysis was inductive and involved line-by-line coding with codes deriving from narratives. A three-step coding procedure was employed [36]. In the first step, the words used by the participants were used to generate meanings. In step 2, data were aggregated to identify the emerging codes and categories. In step 3, theoretical coding was applied to explore relationships between categories [36]. All interviews were double-coded, and the researchers discussed the codes and their definitions.

## 3. Results

Four main themes emerged from the interviews: (1) improvement in the mother–child relationship; (2) a process of personal change during the educative path; (3) the rebuilding of trust relationships; and (4) attitudes and hopes toward the future.

### 3.1. Improvement in the Mother–Child Relationship

One of the most relevant themes in participants’ narratives involved considerations and reflections about their mother–child relationship. In general, women believed that the educative path in the Center allowed them to strengthen and improve the bond with their children. Indeed, after harming experiences of IPV and abuse, during their stay in the Center women had the chance to focus only on their toddlers, neglecting past difficulties and becoming more aware of the relationship with their children.

“*Here, when you are in a mother–child living center, you are only with her, you have just to think about her, so the bond develops more in depth. I see the difference with my other kids…I had many things to think about. Instead, here I have tried to take the best of this path and I am fully enjoying my baby*” (A.).


*“Here, we have many activities to share with the children, can spend good time with them. You know, when you come from a harmful condition, as I do, you really need to create a good relationship with your child, because he has seen awful scenes and can put a blame on you. Spending time with children can help them in trusting us again. This path is helpful in this sense”*
(G.F.).

Furthermore, time spent together increased women’s ability of properly identifying and addressing children’s needs. The participants recognized that playing together is essential for the relationship with their toddlers and for the well-being of both.


*“Since you are alone with her, you immediately feel every need she has, because you devote yourself just to her for 24 hours”*
(A.).


*“When you play with babies, you enter their world. That play is their life in that moment. They talk to you. I can tell them a story. And I enter their world, and in this way I understand if a child is sad or happy”*
(D.).


*“Before, when my child called me to play with him, I used to answer ‘play alone, mam has many things to do’, I didn’t pay attention to his requests. Now I do. You know, he asks you so many questions, he tells you about the school and he really likes it when you are there just to listen. This is a big change the Center helped me to make and I am aware of it”*
(L.).

The improvement of the mother–child relationship was also favored by the development of specific competences, such as encouraging communication, listening, and being influential in giving rules. Indeed, participants reported that in the past, they tended to show either aggressiveness or weakness. Educators are recognized as supportive teachers in the promotion of such abilities.


*“I used to say ‘Yes’, always ‘Yes’, and so my kids had no rules. The educators taught me to give them few simple rules: listen, don’t fight, play together. Normal rules kids are usually given”*
(G.F.).


*“I have learnt that the child should not be excluded and neglected. If kids understand that you really listen to them, they will grow up with this idea. And one day, if something is wrong, they’ll know that they can rely on their mother, they can tell her their problems”*
(L.).


*“Sometimes, we bicker, and he doesn’t want to talk to me. Thanks to the educators, I go there and really try to have him talk because I know that he has a desperate need to communicate. I know that the past was not easy, even for him”*
(N.).


*“After this path, I’ve learnt that you have to be patient when a child cries or has a tantrum. You have to ask: ‘What’s wrong? Are you hungry? Are you sad? Why are you crying?’. I didn’t know that before. I asked just once ‘why are you crying?’ and stopped. Now I am more patient with my children and I ask them until I understand why they are crying o are having a tantrum in that moment”*
(D.).

### 3.2. The Process of Personal Change

Personal change is another key topic. Most women feel that their stay in the Center has been a period of personal growth that has led them to take their own responsibilities as women and mothers. Furthermore, they started exploring their personal characteristics and developing a new image of themselves, after years of abuse and violence.


*“I’ve grown so much, with the educators’ help! Once I got in the Center, I started to know myself. I didn’t know anything about me, in the past I was shown only my negative features. I’ve familiarized with myself…before I couldn’t see me. This is a very big change”*
(A.R.).


*“I’ve changed many things… I deal with problems with more resolution, I carry out the commitments I make…I feel changed, I am more mature, let’s say. I can manage my situation as a whole”*
(A.).


*“They [the educators] have shown me the importance of being a mother, of being a woman. If a man hurts you, you don’t have to completely humiliate yourself”*
(G.F.).


*“About my limitations and insecurity, I’ve understood that I can make it. I will make it once I will be out of this Center. I don’t need anyone, just someone to show me the way”*
(D.).

Their change is strictly connected with their problematic past experiences with partners.


*“The Center helped me knowing many aspects of G. that I didn’t know. I mean…I grew up without a real family, I was alone, in a boarding school. I’ve always hoped something would change with G., but it didn’t happen…this path made me reason about who G. really is”*
(G. F.).


*“I lived my life in fear. Living with fear is not easy at all. I had no self-esteem. Now, thanks to the love I was given here, I’ve regained many things I missed. Now I’m fine”*
(N.).


*“I would make other choices than those I made in the past… I was not strong enough with him, my self-esteem was too low, and I endured too much… if I could get back to the past, I would run away sooner from my situation and would protect my children, much more…but only now I can understand this kind of stuff…”*
(G.).

*“When I fell in love with my husband, I thought he was more capable than me. Then I realized it was untrue. I was pregnant and had to go to the soup kitchen because he didn’t want to work. The house was cold, and we had no hot water. Oh no, If I could go back, I would escape sooner*” (A.R.).

### 3.3. The Rebuilding of Trust Relationships

Six women reported that the educative path in the Center helped them to adopt more appropriate behaviors toward others, and this, in turn, led to the development of enjoyable social relationships.


*“You know, we come from different countries, we have different thought, and cultures, we have to learn to live together. And the Center has helped me to live social relationships in a better way. I have improved my relations with others”*
(G.).


*“I don’t get angry so fast, now. I try to understand people. We are all humans. I‘ve changed my attitude here and I try to have the same attitude also with people outside the Center: at school, at the office, on the bus… I am patient, and my relationships are better”*
(D.).


*“I had the chance to develop good relationships with other mums here… At the beginning, when other mums did some activities with their children, they tried to involve me. They said ‘Come on, this shall pass, you’ll see’. And they were right: by continuing to persist, things have changed, and I am grateful to them for their invitation”*
(L.).

On the contrary, two participants were critical toward their relationships within the Center.


*“I don’t think that living here, with other women, has helped me in improving my attitudes toward others. I did all by myself. Many mums just tell you their story because they want to pour out. But they are not really interested in you”*
(A.).


*“There’s no friend here. If you tell someone something, it’s just because you don’t have any other people to talk to”*
(J.).

When women had a positive experience concerning social relationships, they could also rebuild their primary bonds with family and friends.


*“I was a girl who left everything for her husband. I was completely dependent on him. I had no friends. Today, after more than one year, I have all that I desire: male friends, female friends, freedom…all the relations that I hadn’t before”*
(G.F.).


*“Everytime I went out of the Center, I was afraid to meet people who hurt me. Educators have helped me so much! They told me to remain near the Center and to avoid my old areas. Today I go out and I have no problem. I have found again my sister and my family. I can spend time with them, and this makes me feel strong. Now I know that when my sister scolds me it’s because she loves me. Now I know that to be happy I need my family”*
(N.).

### 3.4. Attitudes and Hopes Toward the Future

Projecting themselves in the future represents one of the most frequent issue in women’s narratives. All the participants appear to share the hope of a stable and serene family.


*“Having a good family, passing down to my children important values, being a responsible mum…not different, but more responsible. I can’t wait! And with the work done here, I can make it”*
(A.).


*“Today, my main project is creating a stable family. I want my child to have stability, both familiar and school stability. Yes, a secure family environment”*
(L.).

Another central plan for the future is related to find a job and live a “typical” everyday life.


*“My goals are: finding a job, having a house, waking up every morning to take him to the nursery school. Living as normal people who don’t live in a Center”*
(J.).


*“Now my future is with a job and with my children. I hope I will find a job soon”*
(G.F.).

Having an occupation is essential to be independent and autonomous. Financial stability would allow participants to satisfy their children’s and their own needs and to protect themselves in case of future difficulties.


*“The job is important if you are a single mum. You have a lot of expenses, and having a job is necessary and useful. I think you should have a job even if you are in a relationship: being self-sufficient is fundamental”*
(A.).


*“Next week, I’ll go back to school. In Nigeria I had school degrees, but they are not valid here in Italy. Finding a job is very important and can completely change my life. I can pay for my child’s school. If he needs an exercise book, I can buy it”*
(J.).


*“I am alone in this country. I am receiving support, food, a house. But having a job could really change my life. For my children’s needs, for me as well. With a job, I could live with them and be autonomous. And I could save some money for them, for their future”*
(G.).

Sharing plans with professionals is a useful strategy to cope with the fear of the future.


*“I’ve thought about my future so many times! And I have talked with educators here, of my plans, my expectations, my fears…it was effective. We’ve planned what I can do when I will go out… I can cook, I like cleaning, looking after children. I’ve realized that I can be a mum and at the same time I can have a job that I like”*
(D.).


*“I shared with the educators my plans and they have supported me. Talking with them, I’ve understood that I would like to work in a coffee shop. I know that when there’s something wrong, I feel better if I prepare meals and coffees. So, I’ve decided to do something new”*
(G.F.).


*“[The professionals] help you understand what it’s better for you, for your future and that of your child. You may know many things, but perhaps you don’t see what it’s better for you. They are essential to show you what it’s right, how you can start again”*
(L.).

## 4. Discussion

The present study aimed at investigating the experience of victims of IPV engaged in an educative path in a Mother–Child Assisted Living Center, with a particular focus on potential indicators of resilience.

The first significant topic in women’s narratives referred to the relationship with their children. In their perception, such relationship was greatly improved during their stay in the Center, as they had the chance to focus exclusively on their toddlers. In other words, they could overcome the “spillover hypothesis” [15,16]. As seen, IPV effects extend beyond the partners relationship and may hinder good parenting by fostering stress and dysfunctional interactions. When women are given the opportunity to feel safe and free from stress and risky behaviors connected to IPV, they can spend their time and energy on building a good mother–child relationship. Consistent with this argument, participants reported that they developed new skills, such as encouraging communication with and listening to their children, and became more able in identifying and addressing their children’s needs. As the literature has largely shown [22,23], exposure to IPV is negatively linked to child development also because of the low levels of children’s stimulation by their mother. 

Thus, if exposure to IPV creates painful experiences that decrease positive emotions about motherhood and disturb an appropriate mother–child bonding [19,20,21], the engagement in an educational path in the Center allows to recuperate positive attitudes about motherhood and develop a more functional relationship with the children. Furthermore, the participants started a process of personal change, developing a new perception of themselves not only as mothers, but also as individuals. In line with the few studies about resilience in IPV victims [30,32], our findings reveal the importance of internal changes, determination, self-pride, and new self-attitudes.

This personal change involves also the representation of the future. Reflections and hopes toward the future represented a key aspect in women’s narratives. Expectations concerned two main life domains, i.e., family and job. The participants dreamed of a ‘normal’ family life, with a serene and peaceful routine for themselves and their children, and a job represented the best way to acquire autonomy and independence, in order to grow the kids in such a positive environment. The professional help received by the educators was explicitly underlined in reference to planning the future. Indeed, professional support resulted to be crucial to helping women to cope with fears and to project themselves in the time to come. As Choi and colleagues noted [37], interventions by professionals are vital to overcome both the challenges and the risks of re-victimization during this period.

Professionals and other women were decisive also for experiencing positive relationships within the Center and rebuilding important bonds with family and friends. In other words, the participants learned to trust different sources of social support and became aware that such support was highly important in their everyday life. Indeed, the literature has clearly shown that effective social-support systems can strengthen IPV victims’ resilience, empowering them to successfully cope with stressful and challenging situations [37].

The present paper presents some limitations which suggest directions for future research. First, the participants were women with a positive experience of staying in the Centers. Only two subjects reported a critical issue, i.e., the absence of good relationships with other hosts, but the narratives were generally favorable. If it is important to understand the successful process of resilience, involving women with mainly negative experiences could greatly contribute to addressing interventions also in case of failure. Second, our findings were based on mothers’ self-reports, which may be influenced by response biases due to the sensitive nature of the topics and to the negative outcomes of full disclosure. Third, as seen, studies on victims of IPV engaged in an educative path are very scarce. However, our sample size was small. Although the qualitative approach we chose allows reaching a more in-depth comprehension of experiences, future research should consider recruiting a larger number of women. Finally, all participants recruited for this study came from Italy. Since the cultural dimension appears to play a key role on both IPV and supportive educative paths [38], cross-cultural investigations should assess the replicability of these findings.

## 5. Conclusions

Victims of IPV who participated in this study reported four main issues about their educative path in a Mother–Child Assisted Living Center: (1) improvement in the mother–child relationship; (2) a process of personal change; (3) the rebuilding of trust relationships; and (4) attitudes and hopes toward the future. Taken together, these main themes emerged from women’s narratives are indicators of the process of resilience, conceived from a socioecological perspective [29] as the ability to use resources rooted in interconnected systems to overcome adversity. In fact, during their engagement in the path, our participants became able to overcome adversity by relying on personal factors, such as internal changes, and on social and institutional networks, such as their family and friends, as well as the Center’s professionals. Another key aspect of the path was the improvement or establishment of an effective mother–child bonding.

The findings of the present study have several implications for our understanding of IPV victims’ experience of resilience. The engagement in an educative path such as the one examined can be very helpful to recuperate positive attitudes toward oneself and motherhood and to overcome the risk of re-victimization. Consistently, it is likely to also promote the healthy development of young children. Although, in the present study, children’s health and well-being were not directly considered, we may suppose that the changes and skills reported by the mothers would improve the life trajectory of their children. Thus, this kind of intervention should be considered as a primary option by professionals working with abused women. Furthermore, policy-makers and community-funding organizations should consider providing adequate resources for developing this service as a standard support for all victims of IPV, both mothers and women without children.

## Figures and Tables

**Table 1 ijerph-17-08318-t001:** Characteristics of the participants.

Participant	Age	Country of Origin	Children (Age)	Educative Path Duration
A.	38	Italy	3 (1, 16, 21 years.)	13 months
D.	33	Nigeria	2 (2, 4 years)	23 months
G.	32	Peru	4 (1, 5, 8, 15 years)	14 months
J.	19	Nigeria	1 (2 years)	2 years
G.F.	29	Italy	2 (4, 5 years)	15 months
L.	25	Italy	1 (5 years)	1 year
N.	42	Morocco	3 (9, 16, 20 years)	14 months
A.R.	32	Romania	1 (4 years)	3 years

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
