# Peer review of "Intimate Partner Violence and Resilience: The Experience of Women in Mother-Child Assisted Living Centers"

_ijerph, 2020, doi:10.3390/ijerph17228318_

Round 1

Reviewer 1 Report

Summary of the Manuscript:

This study investigates the experiences of 8 women living in Mother-Child Assisted Living Centers following experiences of interpersonal violence. The aim of the study is to identify factors that may promote resilience among these women. The qualitative interviews suggested a pattern of four different themes: 1. Improvement in the mother-child relationship; 2. Process of personal change during the educative path; 3. Rebuilding of trusting relationships; and 4. Attitudes and hopes toward the future.

Feedback for the Author(s): 

The study focuses on an important and underexamined population of mothers that face significant adversity and health disparities. This topic has important public health implications, such that interventions and efforts to promote resilience have the potential to improve the life trajectories of both mothers and children. With limited existing research on resilience in this population, the current study is warranted in investigating the experiences of women recovering from IPV to identify factors that would promote resilience and help them to overcome adversity. Overall, the manuscript is well-written and organized, and provides timely insight to the needs of a vulnerable population.  Still, the authors are encouraged to address several concerns to improve and strengthen the manuscript.                                                      

  • Although the authors report that the women were recruited through the Centers, it is not clear whether these women continue to live in the Centers, or whether these women have completed their programs and reintegrated into society. Based on the quotes from participants, it seems that many (all?) of the women are still living in the Centers.
  • It would be helpful to know the average duration of stay for women in the Centers generally, and the rates of women completing the programs. Is it meaningful that these women have remained in the Center for 1-3 years?
  • There is some confusion about how resilience is being defined and conceptualized. The notion of resilience discussed in the introduction seems to focus on gaining autonomy and independence, and overcoming adversity. If the focus of the study is on factors that “strengthen resilience,” it is important to understand the ways that these women are considered resilient. Or is it that these factors themselves are the indicators of resilience? This needs clarification. If the women are still living in the Centers, it would seem that greater emphasis might be given in the introduction to the journey or process or pathway to resilience, and what factors are keeping these women going.
  • It would be helpful for the authors to provide some context about the interview prompts to get a sense of what specifically the participants were asked.
  • The conclusion is very limited and does not provide an adequate summary of the aims of the study or the key findings. The final sentence suggests that effective mother-child bonding is an outcome stemming from the other resilience-strengthening factors, but it is not clear how the authors came to this conclusion based on the presented results. Greater clarification and justification for this claim needs to included.
  • The authors also conclude that the findings “highlight the process of resilience…as the ability to use resources rooted in interconnected systems.” This does not clearly relate to the findings of the study, as the definition/conceptualization of resilience needs to be clarified from the beginning. Is “engagement in an educational path” and sticking with it the marker of resilience, or are these 4 themes the markers of resilience?
  • The authors do not discuss any implications of the findings, or how the information benefits the field. This is important for helping the reader understand how this information can be utilized to help women and families.

Thank you for the opportunity to review this manuscript. The findings provide an insightful view into the experiences faced by abused mothers in Mother-Child Assisted Living Centers. Overall, the study has the potential to make an interesting contribution to the literature, but additional clarification on the aims of the study and how resilience is being defined/understood/conceptualized is needed, as well as how this information can be helpful to the field.

Author Response

We would like to thank the Reviewers for their helpful and supportive comments. We have addressed each comment. Please find below our point-by-point explanation of the revisions in the manuscript.

Please note that we used the “track changes” function to make changes easily visible.

REVIEWER 1

Comments and Suggestions for Authors

Summary of the Manuscript:

This study investigates the experiences of 8 women living in Mother-Child Assisted Living Centers following experiences of interpersonal violence. The aim of the study is to identify factors that may promote resilience among these women. The qualitative interviews suggested a pattern of four different themes: 1. Improvement in the mother-child relationship; 2. Process of personal change during the educative path; 3. Rebuilding of trusting relationships; and 4. Attitudes and hopes toward the future.

Feedback for the Author(s): 

The study focuses on an important and underexamined population of mothers that face significant adversity and health disparities. This topic has important public health implications, such that interventions and efforts to promote resilience have the potential to improve the life trajectories of both mothers and children. With limited existing research on resilience in this population, the current study is warranted in investigating the experiences of women recovering from IPV to identify factors that would promote resilience and help them to overcome adversity. Overall, the manuscript is well-written and organized, and provides timely insight to the needs of a vulnerable population.  Still, the authors are encouraged to address several concerns to improve and strengthen the manuscript.                                                 

  • Although the authors report that the women were recruited through the Centers, it is not clear whether these women continue to live in the Centers, or whether these women have completed their programs and reintegrated into society. Based on the quotes from participants, it seems that many (all?) of the women are still living in the Centers.

Actually, this datum was not reported. We have added it in the Participants section.

  • It would be helpful to know the average duration of stay for women in the Centers generally, and the rates of women completing the programs. Is it meaningful that these women have remained in the Center for 1-3 years?

In the description of the Centers we have added the information about the average duration of stay. Unfortunately, we have not found data about the rates of women completing the programs.

Yes, our women have remained in the Center for 1-3 years, as reported in Table 1.

  • There is some confusion about how resilience is being defined and conceptualized. The notion of resilience discussed in the introduction seems to focus on gaining autonomy and independence, and overcoming adversity. If the focus of the study is on factors that “strengthen resilience,” it is important to understand the ways that these women are considered resilient. Or is it that these factors themselves are the indicators of resilience? This needs clarification. If the women are still living in the Centers, it would seem that greater emphasis might be given in the introduction to the journey or process or pathway to resilience, and what factors are keeping these women going.

Thanks to this “external” point of view we have realized that there was actually some confusion on the concept of resilience. In line with the definition of resilience reported in the paragraph 1.1, we meant to explore indicators of resilience. We have changed some sentences in The current study paragraph to clarify this point. We have removed references to “factors that strengthen resilience” and we have referred to indicators of resilience during women’s path. We have changed the first sentence of the Discussion accordingly.

  • It would be helpful for the authors to provide some context about the interview prompts to get a sense of what specifically the participants were asked.

In the paragraph 2.2 we have added the most frequent prompt questions we used.

  • The conclusion is very limited and does not provide an adequate summary of the aims of the study or the key findings. The final sentence suggests that effective mother-child bonding is an outcome stemming from the other resilience-strengthening factors, but it is not clear how the authors came to this conclusion based on the presented results. Greater clarification and justification for this claim needs to included.

We have revised the Conclusions section. We have added a summary of the main findings and we have changed the sentence about the mother-child bonding. Actually, it was not an outcome, but another key point underlined by participants.

  • The authors also conclude that the findings “highlight the process of resilience…as the ability to use resources rooted in interconnected systems.” This does not clearly relate to the findings of the study, as the definition/conceptualization of resilience needs to be clarified from the beginning. Is “engagement in an educational path” and sticking with it the marker of resilience, or are these 4 themes the markers of resilience?

We have changed some sentences in the Conclusions section to clarify that the main themes are considered indicators of resilience (in line with the present version of The current study paragraph).

  • The authors do not discuss any implications of the findings, or how the information benefits the field. This is important for helping the reader understand how this information can be utilized to help women and families.

We have added implications at the end of the Conclusions paragraph.

Thank you for the opportunity to review this manuscript. The findings provide an insightful view into the experiences faced by abused mothers in Mother-Child Assisted Living Centers. Overall, the study has the potential to make an interesting contribution to the literature, but additional clarification on the aims of the study and how resilience is being defined/understood/conceptualized is needed, as well as how this information can be helpful to the field.

Reviewer 2 Report

The manuscript Intimate Partner Violence and Resilience: The Experience of Women in Mother-Child Assisted Living Centers explores womens experience in safe center envioronment and analysis womens resilience. 

In the introductory part of the manuscript it would be important to elaborate why qualitative research was needed. And in the part of the current study to explain how authors came to the research idea and main question. The information about center would be better to move to methodology part- as setting information.

In the method part the information how interview was constructed, what kind of questions participants were asked lacked. Maybe information about how interpretations were made would clarify better the results. What kind of qualitative methodology was used- it seems like thematic analysis, but authors do not explain their choise at all. It would also important to understand how authors are related with this center and how they tried to ensure the reliability and validity of the results- did authors had a diary or their own reflection or etc.

The biggest issue is the result part. The main themes are too broad and sound as cliche little bit. I would suggest to re- think how main themes are presented and try to reveal what these women experience may mean.

Author Response

We would like to thank the Reviewers for their helpful and supportive comments. We have addressed each comment. Please find below our point-by-point explanation of the revisions in the manuscript.

Please note that we used the “track changes” function to make changes easily visible.

REVIEWER 2

Comments and Suggestions for Authors

The manuscript Intimate Partner Violence and Resilience: The Experience of Women in Mother-Child Assisted Living Centers explores womens experience in safe center envioronment and analysis womens resilience. 

In the introductory part of the manuscript it would be important to elaborate why qualitative research was needed. And in the part of the current study to explain how authors came to the research idea and main question. The information about center would be better to move to methodology part- as setting information.

In the Introduction section we have added a sentence to explain more in depth why qualitative methodology was selected. We have also clarified how we came to the research idea. We have moved the description of the Centers to the Materials and Methods section.

In the method part the information how interview was constructed, what kind of questions participants were asked lacked. Maybe information about how interpretations were made would clarify better the results. What kind of qualitative methodology was used- it seems like thematic analysis, but authors do not explain their choise at all. It would also important to understand how authors are related with this center and how they tried to ensure the reliability and validity of the results- did authors had a diary or their own reflection or etc.

We have added information about the construction and pre-test of the questions and have reported the specific prompt questions we used. We have explained the choice of the thematic analysis, referring explicitly to Corbin and Strauss’ guidelines. We have also added some information about the author who carried out the interviews.

The biggest issue is the result part. The main themes are too broad and sound as cliche little bit. I would suggest to re- think how main themes are presented and try to reveal what these women experience may mean.

Since on this point Reviewers disagree (according to other evaluations our results were clearly and appropriately presented), we did not change the main themes. However, we have substantially revised the Conclusions section to better clarify the key findings focusing on what these women experience may mean.

Round 2

Reviewer 1 Report

Feedback for the Author(s):

 Thank you for the opportunity to review this revised manuscript. The authors were responsive and thorough in their responses to reviewer comments and manuscript revisions. The revised manuscript more clearly specifies the methodology and procedures, and the adjustments to wording in the introduction and discuss help to clarify that the aim of the study is to examine indicators of resilience. The conclusion is improved, and the implications provide a better understanding of the importance of the study. Overall, the study makes an interesting contribution to the field in examining qualitative indicators of the process of resilience among mothers engaging in educative paths following IPV.

My additional comments are very brief:

The authors begin referring to the “educative path” as just “path”, and it reads a bit odd. For example, in section 1.2, authors added, “…with a specific focus on indicators of resilience during such path.” The use of the word path is not clear/correct in this clause. It would be better to use “..during such educative paths” or “…”with a specific focus on indicators of resilience that arise during their time in the Centers.”

Similarly, the authors refer to “the path” in the conclusion, and editing it to say “…engagement in their educative paths…” would clarify the meaning. Please address other instances of the use of “path” in the conclusion.

In the conclusion, the authors might consider changing the wording of “..reported four main issues…” to “…reflected on four main themes…”, as the word issue has a negative connotation suggesting a problem.

The sentence, “Taken together, these main themes emerged from women’s narratives are indicators of the process of resilience…” needs to be edited to be clearer. The grammar is not quite correct.

One thing that still seems to be missing in the conclusion is making the link between the themes (as indicators of resilience or the process of resilience) and how they are actually indicating or leading to “overcoming adversity”. The authors suggest they are part of the process of overcoming adversity, but what does “overcoming adversity” look like for these women? Additionally, although the sentence referring to mother-child bonding is clarified as an indicator of resilience, it is not well integrated into the conclusion and makes for an odd ending to the paragraph.

In section on implications, the word victims should have an apostrophe to indicate possessive, E.g., “…our understanding of IPV victims’ experience of resilience.”

Thank you for your efforts with this revision. 

Author Response

REVIEWER 1

We would like to thank the Reviewer for the opportunity to improve our manuscript. His/her comments were particularly helpful and accurate.

Feedback for the Author(s): Thank you for the opportunity to review this revised manuscript. The authors were responsive and thorough in their responses to reviewer comments and manuscript revisions. The revised manuscript more clearly specifies the methodology and procedures, and the adjustments to wording in the introduction and discuss help to clarify that the aim of the study is to examine indicators of resilience. The conclusion is improved, and the implications provide a better understanding of the importance of the study. Overall, the study makes an interesting contribution to the field in examining qualitative indicators of the process of resilience among mothers engaging in educative paths following IPV.

My additional comments are very brief:

The authors begin referring to the “educative path” as just “path”, and it reads a bit odd. For example, in section 1.2, authors added, “…with a specific focus on indicators of resilience during such path.” The use of the word path is not clear/correct in this clause. It would be better to use “..during such educative paths” or “…”with a specific focus on indicators of resilience that arise during their time in the Centers.”

Similarly, the authors refer to “the path” in the conclusion, and editing it to say “…engagement in their educative paths…” would clarify the meaning. Please address other instances of the use of “path” in the conclusion.

We have changed the use of “path” into “educative path” in all the sections of the paper.

In the conclusion, the authors might consider changing the wording of “..reported four main issues…” to “…reflected on four main themes…”, as the word issue has a negative connotation suggesting a problem.

We have changed this sentence in line with the suggestion.

The sentence, “Taken together, these main themes emerged from women’s narratives are indicators of the process of resilience…” needs to be edited to be clearer. The grammar is not quite correct.

We have simplified and clarified the sentence.

One thing that still seems to be missing in the conclusion is making the link between the themes (as indicators of resilience or the process of resilience) and how they are actually indicating or leading to “overcoming adversity”. The authors suggest they are part of the process of overcoming adversity, but what does “overcoming adversity” look like for these women? Additionally, although the sentence referring to mother-child bonding is clarified as an indicator of resilience, it is not well integrated into the conclusion and makes for an odd ending to the paragraph.

We have specified that overcoming adversity can be the ability to cope with their past experience of abused women and present experience of mothers. This reference to motherhood can also clarify the link with the sentence about the mother-child bonding.

In section on implications, the word victims should have an apostrophe to indicate possessive, E.g., “…our understanding of IPV victims’ experience of resilience.”

We have added the apostrophe.

Reviewer 2 Report

Thank you for the corrections you have made. I think the manuscript improved, though it would be good to mention in the abstract exat method which was used for data analysis.

Author Response

REVIEWER 2

We would like to thank the Reviewer for the opportunity to improve our manuscript. His/her comments were particularly helpful.

Thank you for the corrections you have made. I think the manuscript improved, though it would be good to mention in the abstract exat method which was used for data analysis.

In the Abstract we have added information about data analysis methodology.